# Fabrication, Structural Properties, and Electrical Characterization of Polymer Nanocomposite Materials for Dielectric Applications

**DOI:** 10.3390/polym15143067

**Published:** 2023-07-17

**Authors:** Ali Atta, Mohammed F. Alotiby, Nuha Al-Harbi, Mohamed R. El-Aassar, Mohamed A. M. Uosif, Mohamed Rabia

**Affiliations:** 1Physics Department, College of Science, Jouf University, Sakaka P.O. Box 2014, Saudi Arabia; mauosif@ju.edu.sa; 2Nuclear Technologies Institute (NTI), King Abdulaziz City for Science & Technology (KACST), P.O. Box 6086, Riyadh 11442, Saudi Arabia; mfotiby@kacst.edu.sa; 3Department of Physics, Faculty of Applied Sciences, Umm Al-Qura University, Makkah, Saudi Arabia; 4Chemistry Department, College of Science, Jouf University, Sakaka P.O. Box 2014, Saudi Arabia; mrelaassar@ju.edu.sa; 5Nanomaterials Science Research Laboratory, Chemistry Department, Faculty of Science, Beni-Suef University, Beni-Suef 62514, Egypt; mohamedchem@science.bsu.edu.eg; 6Nanophotonics and Applications Laboratory, Physics Department, Faculty of Science, Beni-Suef University, Beni-Suef 62514, Egypt

**Keywords:** fabrication, structural, electrical, nanocomposites, energy applications

## Abstract

This research paper aims to fabricate flexible PVA/Cs/TiO_2_ nanocomposite films consisting of polyvinyl alcohol (PVA), chitosan (Cs), and titanium oxide (TiO_2_) for application in energy storage devices. The samples were analyzed using X-ray diffraction (XRD), atomic force microscope (AFM), scanning electron microscopy (SEM), Fourier transform infrared spectroscopy (FTIR), and energy dispersive X-ray (EDX) techniques. The impact of TiO_2_ on the electrical impedance, conductivity, permittivity, and energy efficiency of the PVA/Cs was determined in a frequency range of 100 Hz to 5 GHz. The XRD, FTIR, and EDX results showed the successful fabrications of the PVA/Cs/TiO_2_. The SEM and AFM images illustrated that the TiO_2_ was loaded and distributed homogenously in PVA/Cs chains. In addition, the electrical conductivity was enhanced from 0.04 × 10^−7^ S.cm^−1^ of PVA/Cs to 0.25 × 10^−7^ S.cm^−1^ and 5.75 × 10^−7^ S.cm^−1^, respectively, for the composite PVA/Cs/0.01TiO_2_ and PVA/Cs/0.1TiO_2_, and the dielectric constant grew from 2.46 for PVA/Cs to 7.38 and 11.93, respectively. These results revealed that modifications were made to the produced films, paving the way for using the composite PVA/Cs/TiO_2_ films in different energy applications, such as electronic circuits and supercapacitors.

## 1. Introduction

Nanocomposites are getting a lot of focus in storage energy due to their exceptional mechanical and electrical characteristics [1]. Polymers were used in a wide variety of storage devices thanks to the addition of inorganic filler, which enhances their structural features and mechanical capabilities [2,3]. Researchers are seeking to enhance the properties of polymer nanocomposites for practical applications in the manufacturing sector [4]. Polymer nanocomposites attract interest in several alternative technical applications due to their ease of use and production [5]. The need to produce polymer nanocomposites with varied characteristics for energy applications was emphasized in several recent literature reviews [6,7].

PVA/Cs/TiO_2_ composite is a promising material with a wide variety of applications due to its properties of biodegradable, mechanical, optical characteristics, and thermal stability. PVA/chitosan are frequently discussed in the literature. For instance, Wu et al. [8] created films from PVA/chitosan blends and hypothesized that the two materials interacted via physical crosslinks mediated by the creation of intermolecular hydrogen bonds. There is a synergistic effect on many properties of PVA/chitosan films when nanoparticles are incorporated as a filler as recorded by Jiang et al. [9]. Fan et al. [10] showed that adding chitosan to PVA/Gelatin mixes increased their tensile strength. Based on their investigation of PVA/chitosan blends, Chen et al. [11] determined that chitosan could reduce the crystallinity of PVA at various ratios.

Because of its improved process ability and reversible conductivity control, polyvinyl alcohol (PVA) is one of the polymers with promising applications [12]. PVA is extensively investigated [13] due to its resistance to ultraviolet radiation, lightning-fast reactivity, configurability, and manageability. PVA is a vinyl alcohol-co-vinyl acetate copolymer that is physiologically inert, bio-adhesive, and biodegradable. It can withstand high forces, is gas-permeable, and dissolves in water. Both the UV-vis and dimensional stability of PVA are excellent [14]. The stiffness of blended chitosan and PVA with high mechanical properties is preserved. PVA is a great polymer to combine with chitosan as biocompatible and water-soluble material [15]. However, the physical and chemical characteristics of PVA are limited. Therefore, when the polymer is combined with filler elements is applied to overcome these restrictions and increase its usefulness, especially in industrial settings.

Chitosan (Cs) is a natural polymer that is widely used in the biotechnology industry. Its benefits in life science, including its readily available, biocompatible, and biodegradable nature, drive its increased use [16]. Advantages include adaptability to different end uses via chemical or enzymatic modification of derivatives. Chitosan is one of the most adaptable biomaterials since it has the potential to be used in a wide variety of settings, including medicine, natural sciences, agriculture, and even pollution control [4]. Chitosan can be used in numerous sensing and electrochemical applications due to its high hydrophilicity, as well as the physical and chemical modification options, all to their advantage [17]. The molecular weight and distribution of acetyl groups in the structure of chitosan determine how well it dissolves in an aqueous acidic medium. Architectural topography and surface charge, as well as anti-inflammatory, have led to its widespread use in a variety of different applications [18].

Incorporating nanofillers in the polymer has paved the way for tailoring nanocomposites’ features. Furthermore, the nanofillers conduct linkages inside the polymer chains, improving the polymer matrix’s dielectric properties. The host properties can be modified thanks to the nanofillers embedded in the polymeric chains [19]. Moreover, polymer properties are improved by using fillers made of scattered nanoparticles, which are more effective than micron-sized fillers [20]. Differences in structural properties have a significant impact on electron/hole interactions and, in turn, produce distinct dispersion values that modify the features of inter-band optical transitions [21,22].

Titanium dioxide (TiO_2_) as a conductive filler has shown considerable promise in the devices of electronic chips, supercapacitors, and solar cells. There has been growing concern in recent years about using TiO_2_ in several applications, particularly in detectors and lighter emitters [23]. Thus, TiO_2_ is the most promising material for usage in photovoltaic cells, sensors, and electronic devices [24]. One of the most common approaches to creating TiO_2_ dispersion colloids in different solvents is to include TiO_2_ in PVA/Cs [22]. Additionally, TiO_2_ is widely used in photo-electrochemical and cell air-water purification because it is a powerful redox material [25].

TiO_2_ is also widely known as a powerful redox material with applications in photo-electrochemical and cell air–water purification [26]. Furthermore, the TiO_2_ nanoparticles are more likely to interact in a Cs/PVA matrix because of the presence of groups OH in PVA and NH_2_ in Cs. The mechanical properties of the Cs matrix were improved by the interaction of TiO_2_ via the Cs/TiO_2_ composite [27].

The key objective of this work is to improve the PVA/Chitosan dielectric characteristics by including TiO_2_ in PVA/Chitosan blend. TiO_2_ has a high strength-to-weight ratio and a large surface area compared to its volume, both of which would improve a polymer matrix composite's mechanical properties. TiO_2_ is also widely known as a powerful redox material with applications in photo-electrochemical devices [28]. This study aims to develop composite films made of PVA, Cs, and TiO_2_. Some techniques, such as XRD, FTIR, AFM, EDX, and SEM, were used to characterize the produced films. To use these synthesized films in energy storage devices, the dielectric properties of PVA/Cs /TiO_2_ nanocomposite samples were strengthened.

## 2. Materials and Methods

The chemicals, PVA with a molecular weight of 86,500–90,500 g/mol and hydrolysis of 99.4%, TiO_2_ of 99.9%, and Cs with a degree of deacetylation of 88% and purity of 99.7%, were obtained from Sigma-Aldrich Company. For making the composite, the PVA solution (6% *w*/*v*) was made by dissolving PVA in deionized water at 88 °C for 1.5 h. The Cs solution (6% *w*/*v*) was dissolved in deionized water at 88 °C with agitation for 1.5 h, as discussed before [28]. Overnight, 125 mL of 1.1% *v*/*v* acetic acid solution was stirred into a 1.1% *w*/*v* Cs solution. The polymeric combination was obtained by combining the PVA and Cs solutions and stirring the mixture for 5 h. The produced polymer solution was then combined with TiO_2_NPs of varying concentrations to produce a PVA/Cs/TiO_2_ solution, which was then cross-linked using glutaraldehyde, as described before [29]. At 50 °C, the glutaraldehyde (0.55 mL) was gently added to the polymer solution, which was then thoroughly mixed before being cast in the Petri dish. Using a thickness gauge (Mitutoyo 731) and scanning electron microscopy, the thickness of the composite films was determined in the range of 90 μm.

The structural properties were analyzed using XRD (Shimadzu, XRD 6000, Kyoto, Japan), while the chemical changes were investigated using FTIR (ATI Mattson, Bristol, UK). SEM (JEOL, Tokyo, Japan) and AFM (JEOL, Shimadzu) investigated the morphology and geometric shape of the samples, as well as the surface roughness. In addition, the dielectric characteristics of all films were recorded using an LCR meter (RS-232C interface, Hioki, Japan) operating in frequencies of 100 Hz to 5 GHz.

## 3. Results

Films of PVA/Cs and PVA/Cs/0.1TiO_2_ composite are shown in XRD patterns as in Figure 1. The XRD pattern showed the PVA/Cs polymer was semi-crystalline, with a peak of 19.8°, indicating the formation of PVA. The composite of PVA/Cs/TiO_2_ showed a new diffraction peak of TiO_2_ at 25.6° correlated to (111). In addition, a shift in the peak of PVA/Cs with intensity demonstrated crystallinity due to the interaction of TiO_2_ in the PVA/Cs [30]. The insertion of TiO_2_ nanoparticles caused a shift in the peaks because it altered the electron–hole recombination rate caused by the presence of the defects introduced into the crystal structure. The average crystallite size (D) of TiO_2_ was calculated using the Debye-Scherrer formula by [31]
(1)$$D=\frac{0.94\lambda }{\beta \;\cos\theta }$$
λ represents the X-ray wavelength, θ is the Bragg angle, and β is the FWHM. The crystallite size D is determined in the range of 35.2 nm.

The chemical structure and functional groups of the produced composites are studied with FTIR. The FTIR of pristine PVA/Cs and the composite PVA/Cs//TiO_2_ are given in Figure 2. The broad peaks at 3280 cm^−1^ of PVA/Cs were to −OH groups and at 2922 cm^−1^ for C–H stretching vibrations. The peak at 1635 cm^−1^ was for vibrational C=C. Another peak at ~1415 cm^−1^ and 1075 cm^−1^ referred to vibrations of −OH and C=O, respectively. The band 830 cm^−1^ referred to CH_2_ stretches. Due to the interaction between TiO_2_ and PVA/Cs levels, these peaks were seen in all spectra with a slight shift [32]. Moreover, due to charge transport complexes, PVA/Cs/TiO_2_ composite films had a different intensity than pure PVA/Cs [33]. When nanoparticles were added, these bands’ locations shifted from red to blue, showing a change in the interactions between intermolecular hydrogen bonds and confirming the complexation between the polymer and the nanoparticles. This research demonstrates the existence of intermolecular bands due to the presence of hydrogen bonds between Cs and the hydroxyl group of PVA. The resonance interaction between carbonyl and amino groups due to the entanglement of nitrogen non-bonding electrons causes the composite to have a larger wavenumber than that of pure PVA.

Figure 3 illustrates the formation of TiO_2_ in PVA/Cs as analyzed using EDX mapping. The three constituent elements, i.e., O, C, and Ti, were all detectable and homogenous distributions. As shown in Figure 3, the Ti peaks prove the existence of TiO_2_ nanoparticles on the PVA/Cs. Additionally, the EDX mapping showed the dispersion of Ti elements on the structure as a sign of attachment of TiO_2_ to the structure [34,35]. 

Figure 4a–d depict the AFM images of PVA/Cs and PVA/Cs/TiO_2_ in 3D dimensions. The AFM images of PVA/Cs were smoother, as shown in Figure 4a. Moreover, the AFM images of PVA/Cs/TiO_2_, as shown in Figure 4b–d, were rougher compared to PVA/Cs image. Additionally, the AFM results demonstrated that the TiO_2_ increased the surface roughness from 13 nm for PVA/Cs to 32 nm for PVA/Cs/0.1TiO_2_. The addition of TiO_2_ in PVA/Cs caused an increase in surface roughness, demonstrating the mixing of TiO_2_ in the PVA/Cs blend. Hydrophilic in PVA/Cs blend are efficiently covered by TiO_2_, making PVA/Cs/TiO_2_ composite rougher to be more convenient in different applications. Figure 4b,c display the smooth surface of PVA/Cs/0.025TiO_2_ and PVA/Cs/0.05TiO_2_ compared to PVA/Cs/0.1TiO_2_ (Figure 4d). This displays the increase in roughness that resulted from the effective incorporation of TiO_2_. Based on these findings, TiO_2_ is effectively incorporated as a filler into the PVA/Cs matrix. Therefore, the three TiO_2_ percentages were chosen and incorporated into the PVA/Cs polymeric matrix at this ratio.

Figure 5a–e depicts the morphology of PVA/Cs and PVA/Cs/TiO_2_. Figure 4a of a micrograph of PVA/Cs film reveals a consistent and smooth surface. Images of PVA/Cs films containing 0.01TiO_2_, 0.025TiO_2_, 0.05TiO_2_, and 0.1TiO_2_ are shown in Figure 5b–e. Figure 5b–e depicts the production of granule-shaped TiO_2_ in a PVA/Cs, with an average size of TiO_2_ particles of 97 μm. The formation of tiny agglomerates on the polymer surfaces is attributed to the rising TiO_2_ content in the PVA/Cs blend, causing the non-uniform distribution [36]. However, raising the TiO_2_ concentration from 0.01 to 0.1 altered the spherical form of the nanoparticles. The influence of the dopant TiO_2_ concentration and the synthesis conditions might be responsible for the observed morphological changes. In addition, increasing TiO_2_ concentration in PVA/Cs increased the dispersion of TiO_2_ and provided appropriate interfacial interaction between the PVA/Cs chains and TiO_2_ that PVA/Cs is a competent host matrix for encapsulating TiO_2_ [37].

The components of the dielectric permittivity (ε*), the real dielectric constant ($\varepsilon^{\prime }$), and the imaginary dielectric loss (ε″) are calculated by [38]:(2)$$\varepsilon *=\varepsilon^{\prime }-i\;\varepsilon \prime \prime$$

The $\varepsilon^{\prime }$, which measures how much electricity a substance can store, is provided by [39]:(3)$$\varepsilon \prime =\frac{c.\;d}{\varepsilon_{o}.A}$$
where c is capacitance, t is film thickness, and A is area. The $\varepsilon^{\prime }$ with frequency for PVA/Cs and PVA/Cs/TiO_2_ are shown in Figure 6. At low frequencies, the $\varepsilon^{\prime }$ is reduced rapidly for all films. This occurred because the dipoles had adequate time to experience interfacial polarization at these lower frequencies. Nonetheless, at higher frequencies, the $\varepsilon^{\prime }$ approached a constant value, possibly because dipoles did not have adequate time to align themselves along the tendency of applied fields [40].

The dielectric $\varepsilon^{\prime }$ value of PVA/Cs showed significant improvement following the addition of the TiO_2_. The dielectric constant of pure PVA/Cs at 100 Hz was 2.46 and enhanced to 7.38, 9.07, 9.76, and 11.93, respectively, for PVA/Cs/0.01TiO_2_, PVA/Cs/0.025TiO_2_, PVA/Cs/0.05TiO_2_, and PVA/Cs/0.1TiO_2_, as shown in Table 1. The conductivity and uniform distribution of TiO_2_ in PVA/Cs films contributed significantly to the enhancement of the dielectric constant of the composite PVA/Cs/TiO_2_ films. The interfacial polarizability was also enhanced due to the rise in dipole density caused by adding TiO_2_. As a result, the polarization and dielectric of the PVA/Cs polymer matrix were improved by the presence of these flaws. Atta [41] found this behavior when he produced CuNPs on PET and PTFE using an ion source sputtering approach with varying deposition times. At an applied frequency of 105 Hz, he found that the $\varepsilon^{\prime }$ improved from 0.6 for pristine PET to 1.2 for 25 min Cu/PET, and from 1.9 for PTFE to 2.3 for 25 min Cu/PTFE. Because of their high dielectric constant, polymer nanocomposite films are promising candidates for use in energy storage.

The dielectric loss ($\varepsilon^{\prime \prime })$, which represents the fraction of input energy lost as heat by a medium, is given by [42].
(4)$$\varepsilon^{\prime \prime }=\varepsilon^{\prime }\tan\delta$$

The $\varepsilon^{\prime \prime }$ with frequency for PVA/Cs and PVA/Cs/TiO_2_ is presented in Figure 7. It was shown that the increase in frequency reduced the $\varepsilon^{\prime \prime }$ at lower frequency [43]. However, the reported dielectric loss values tended to remain constant in the high-frequency region [44]. In addition, the effects in the low-frequency area were responsible for this shift in the $\varepsilon^{\prime \prime }$ value. The charges effects, which altered electronic oscillations by raising their frequency, were another reason for varying the $\varepsilon^{\prime \prime }$ with frequency. Moreover, the value of $\varepsilon^{\prime \prime }$ was 2.83 for pure PVA/Cs, increased to 9.61 for PVA/Cs/0.01TiO_2_, and 10.58 for PVA/Cs/0.1TiO_2_. The enhancement of $\varepsilon^{\prime \prime }$ by the addition of TiO_2_ was due to an increase in coupling strength across the grain boundary [45], which resulted from an improvement in the linkage of the PVA/Cs and TiO_2_.

The electrical modulus (M*) is determined by t [46].
(5)$$M*=\frac{1}{\varepsilon *}=M\prime +i\;M\prime \prime$$

The real M′ and imaginary M″ modules are given by [47]:(6)$$M^{\prime }=\frac{\varepsilon^{\prime }}{\varepsilon^{\prime 2}+\varepsilon^{\prime \prime 2}}$$
(7)$$M^{\prime \prime }=\varepsilon^{\prime \prime }/\left(\varepsilon^{\prime 2}+\varepsilon^{\prime \prime 2}\right)$$

Figure 8 displays the dependence of $M^{\prime }$ on the applied frequency for PVA/Cs and PVA/Cs/TiO_2_. In the low-frequency range, as depicted in the image, M′ was very close to zero. At very high frequencies, M′ increased at an exponential rate with frequency. Polymers’ electrical dipoles were strongly attracted to an external electric field in the low-frequency area but much less in the high frequencies zone. Furthermore, the relaxation conductivity of electrode polarization affected the electric modulus [48]. As shown in Figure 7, when TiO_2_ was present, the actual electric modulus M′ decreased. The modulus M′ of pure PVA/Cs was 0.174 and reduced to 0.089 and 0.039, respectively, for PVA/Cs/0.01%TiO_2_ and PVA/Cs/0.1%TiO_2_. The electric modulus permittivity decreased as TiO_2_ concentration rose because of the dipolar contribution of charge carriers.

The M″ with frequency for PVA/Cs and the composite PVA/Cs/TiO_2_ is shown in Figure 9. The permittivity M″ values were observed with increasing at a lower frequency, indicating that the ions were moving by jumping from one location to the next. The observed peak in M″, which indicated the presence of a relaxation process, was changed with the content of TiO_2_. The contribution of dipolar groups to the permittivity decreased as the frequency rose because more dipolar groups became un-orientable at higher frequencies. By increasing TiO_2_, the M″ decreased from 0.101 for the PVA/Cs to 0.073 for PVA/Cs/0.01TiO_2_ and to 0.007 for PVA/Cs/0.1TiO_2_. The addition of TiO_2_ shifted the PVA/Cs relaxation peak, which reduced the relaxation ($\tau_{r}$), as given by the relation [49]:(8)$$\tau_{r}=\frac{1}{\omega_{r}}$$
where $\omega_{r}$ is the angular frequency corresponding to the relaxation peak crest of M″. The relaxation time of PVA/Cs was 4.35 × 10^−7^ s and reduced to 3.31 × 10^−7^ s, 3.14 × 10^−7^ s, 2.74 × 10^−6^ s, and 2.6 × 10^−6^ s, respectively, after incorporating TiO_2_ concentrations by 0.001, 0.025, 0.05, and 0.1. Abdelhamied et al. [50] investigated the value of relaxation time as 3.3 × 10^−5^ s for PVA that changed to 3.25 × 10^−5^ s and 1.9 × 10^−5^ s, respectively, for the PVA/PANI blend and PVA/PANI/Ag composite. The reduction in $\tau_{r}$ was because of the increase in dipole mobility, which made the hopping mechanism easier [51].

The complex impedance Z* is determined by the relation [52]:(9)$$Z*=Z\prime +{\mathrm{iZ}}^{\prime \prime }$$
where Z′ and Z″ are the real and imaginary impedance parts. The relation of Z′ with frequency for pristine PVA/Cs and the composite PVA/Cs/0.01TiO_2_, PVA/Cs/0.025TiO_2_, PVA/Cs/0.05TiO_2_, and PVA/Cs/0.1TiO_2_ are shown in Figure 10. The impedance Z′ value of all films was lowered with frequency in the low-frequency zone and was largely fixed at the higher frequencies because of higher conduction and a stable impedance resulting from a greater number of free charges [53]. Furthermore, the addition of TiO_2_ reduced the Z′ value because TiO_2_ increased conductivity and decreased the film impedance directly. The Z′ was decreased from 2.65 × 10^7^ for the pristine PVA/Cs to 2.07 × 10^7^ and 1.06 × 10^7^, respectively, to PVA/Cs/0.01TiO_2_ and PVA/Cs/0.1TiO_2_. This finding was because TiO_2_ processed the dipole mobility contribution of charge carriers.

Figure 11 displays the relationship between the frequency and Z″ of PVA/Cs/TiO_2_. The Z″ dropped with frequency, as is evident from identical actions of $Z^{\prime }$ with frequency. The decrease in Z″ value with frequency in the lower frequency region was because a faster charge transfer was achieved [54]. The value of Z″ of PVA/Cs was 13 × 10^7,^ which reduced, respectively, to 6.5 × 10^7^ and 2.3 × 10^7^ for the composite PVA/Cs/0.01TiO_2_ and PVA/Cs/0.1TiO_2_. This reduction was because TiO_2_ incorporation induced changes in polarizability and due to the rise in dipole density caused by adding TiO_2_.

The energy density (*U*) is computed using this relation [55]:(10)$$U=\frac{1}{2}\;\varepsilon^{\prime }\varepsilon_{O}\;E^{2}$$

*E* is the electric field, and $\varepsilon_{O}$ is the electric permittivity. The value of *U* at various frequencies for pristine PVA/Cs and the composite PVA/Cs/TiO_2_ are shown in Figure 12. The addition of TiO_2_ increased the energy density from 1.05 × 10^−5^ J/m^3^ for PVA/Cs to 3.16 × 10^−5^ J/m^3^, and 3.89 × 10^−5^ J/m^3^ and 5.12 × 10^−5^ J/m^3^ for PVA/Cs/0.01TiO_2_, PVA/Cs/0.025TiO_2_ and PVA/Cs/0.1TiO_2_. The results demonstrated that TiO_2_ was successfully incorporated into the PVA/Cs chains, which enhanced its dielectric responses and capabilities for energy storage.

The electrical conductivity $\sigma_{ac}$ is given by [56].
(11)$$\sigma_{ac}=\varepsilon_{o}\varepsilon \prime \prime \;\omega$$

Figure 13 shows the change in $\sigma_{ac}$ of PVA/Cs and the composite PVA/Cs/TiO_2_ with frequency. Because the applied electric field increased the mobility of charge carriers, the $\sigma_{ac}$ varied with frequency for all films [57]. In addition, there was a shift in $\sigma_{ac}$ after TiO_2_ incorporation [58]. At 100 Hz, the electrical conductivity of pure PVA/Cs was 0.04 × 10^−7^ S/cm, increased to 0.25 × 10^−7^ S/cm, 0.96 × 10^−7^ S/cm, and reached 5.75 × 10^−7^ S/cm, respectively, by increasing the content of TiO_2_ to 0.01, 0.025, and 0.1 in PVA/Cs, as shown in Table 1. This result was due to polymer chain scission and degradation in the following TiO_2_ [59]. The $\sigma_{ac}$ of PVA/MWCNT that was determined by Atta et al. [60] at 1 MHz was 0.9 × 10^−4^ S/cm for PVA, which improved for PVA/MWCNT to 26 × 10^−5^ S/cm. This result was due to MWCNT-generated charge carriers, which improved conductivity.

The maximum height $W_{m}$ of the potential barrier, denoting the highest amount of needed energy to eject the electron from a site to infinity, is determined using the formula [60]:(12)$$W_{m}=\frac{-4k_{B}T}{m}$$

The m value is given by the slopes of the line ln $(\varepsilon^{\prime \prime }$) vs. ln (ω), as shown in Figure 14, using the formula [61,62]:(13)$$\varepsilon^{\prime \prime }={A\omega }^{m}$$

The value of m changed from 0.098 for PVA/Cs to 0.132, 0.156, 0.112, and 0.133, respectively, for PVA/Cs/0.01TiO_2_, PVA/Cs/0.025TiO_2_, PVA/Cs/0.05TiO_2_, and PVA/Cs/0.1TiO_2_ films. With TiO_2_ concentration, the predicted potential barrier $W_{m}$ changed from 1.07 eV for PVA/Cs, respectively, to 0.79 eV, 0.67, 0.93, and 0.78 for PVA/Cs/0.01TiO_2_, PVA/Cs/0.025TiO_2_, PVA/Cs/0.05TiO_2_, and PVA/Cs/0.1TiO_2_ films. Moreover, the effects of incorporating on crystal structure and defect density of TiO_2_ were responsible for this decrease in potential barrier $W_{m}$. The increase in TiO_2_ in PVA/Cs improved its electrical conductivity by lowering the energy needed for charge carrier hopping.

## 4. Conclusions

The solution casting method is used to create flexible PVA/Cs/TiO_2_ composite films successfully. The films were then analyzed using various methods, such as XRD, FTIR, EDX, SEM, and AFM. The XRD, FTIR, and EDX methods verified that the PVA/Cs/TiO_2_ polymeric composite films were fabricated successfully. Furthermore, the AFM results demonstrated that TiO_2_ increased the surface roughness from 13 nm for PVA/Cs to 32 nm for PVA/Cs/0.1TiO_2_. The SEM analysis illustrated that TiO_2_ was successfully integrated into PVA/Cs. The incorporation of TiO_2_ was shown to improve electrical characteristics. Moreover, the electric modulus and complex impedance were found to be sensitive to changes in TiO_2_ content. When the percentage of TiO_2_ was raised from 0.025 to 0.1, the dielectric dispersion properties were enhanced as a result of the migration of charge carriers. The addition of TiO_2_ increased the energy density from 1.05 × 10^−5^ J/m^3^ for PVA/Cs to 3.16 × 10^−5^ J/m^3^, 3.89 × 10^−5^ J/m^3^, and 5.12 × 10^−5^ J/m^3^ for PVA/Cs/0.01TiO_2_, PVA/Cs/0.025TiO_2_, and PVA/Cs/0.1TiO_2_. The results demonstrated that TiO_2_ was successfully incorporated into the PVA/Cs chains, which enhanced its dielectric responses and capabilities for energy storage. The results of this research open up opportunities for the flexible PVA/Cs/TiO_2_ nanocomposite films to be used in a wide range of potential devices, such as batteries and supercapacitors.

## Figures and Tables

**Figure 1 polymers-15-03067-f001:**
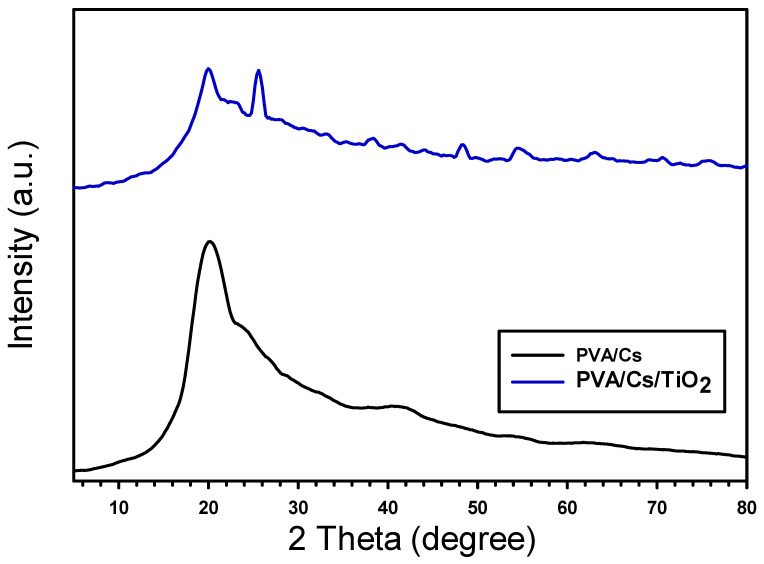
XRD of pristine PVA/Cs and the composite blend PVA/Cs/TiO_2_.

**Figure 2 polymers-15-03067-f002:**
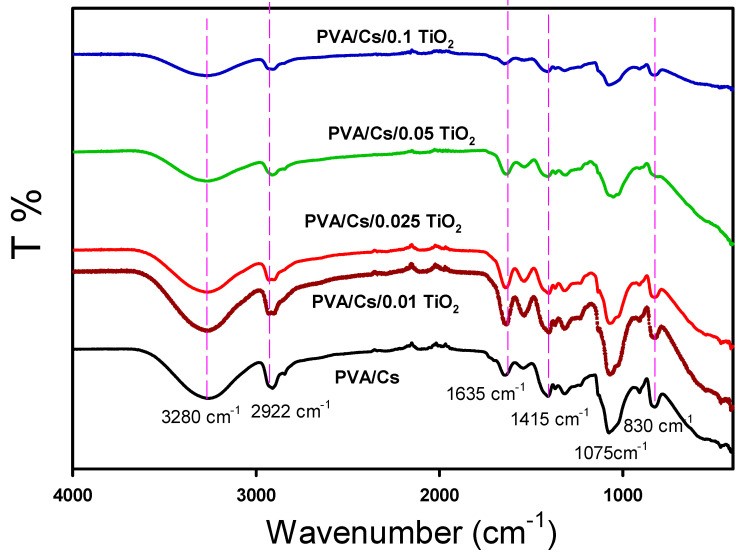
FTIR of the pristine PVA/Cs and the composite PVA/Cs/TiO_2_ films.

**Figure 3 polymers-15-03067-f003:**
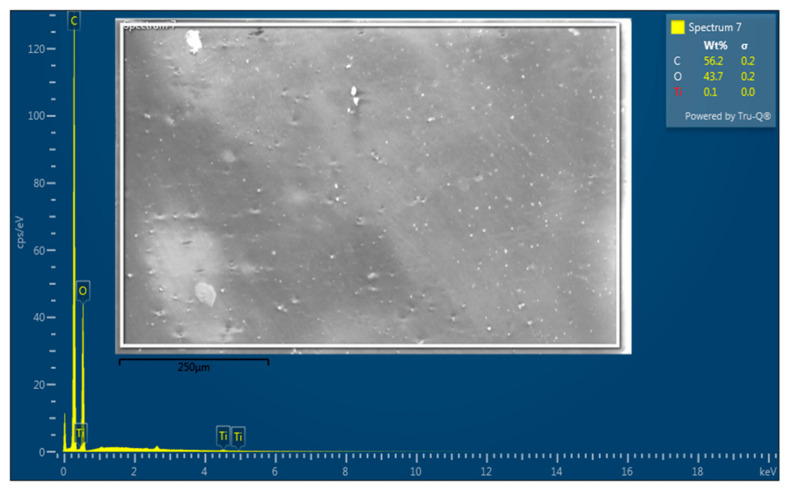
EDX mapping of the PVA/Cs/TiO_2_ composite.

**Figure 4 polymers-15-03067-f004:**
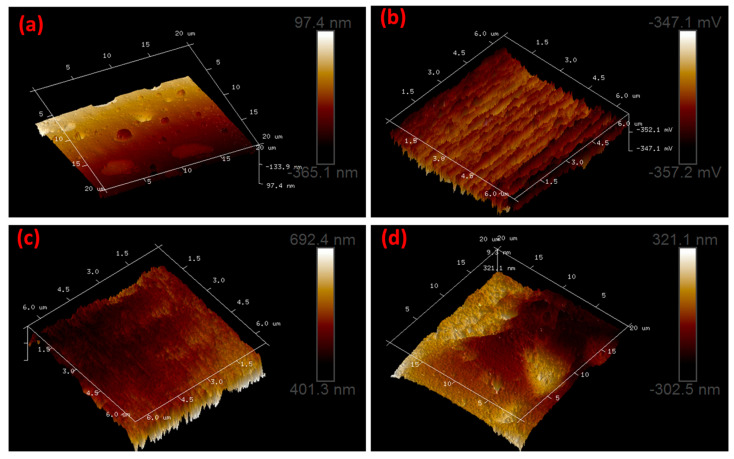
AFM of (**a**) PVA/chitosan, (**b**) PVA/chitosan/0.025TiO_2_, (**c**) PVA/chitosan/0.05TiO_2_, and (**d**) PVA/chitosan/0.1TiO_2_ films.

**Figure 5 polymers-15-03067-f005:**
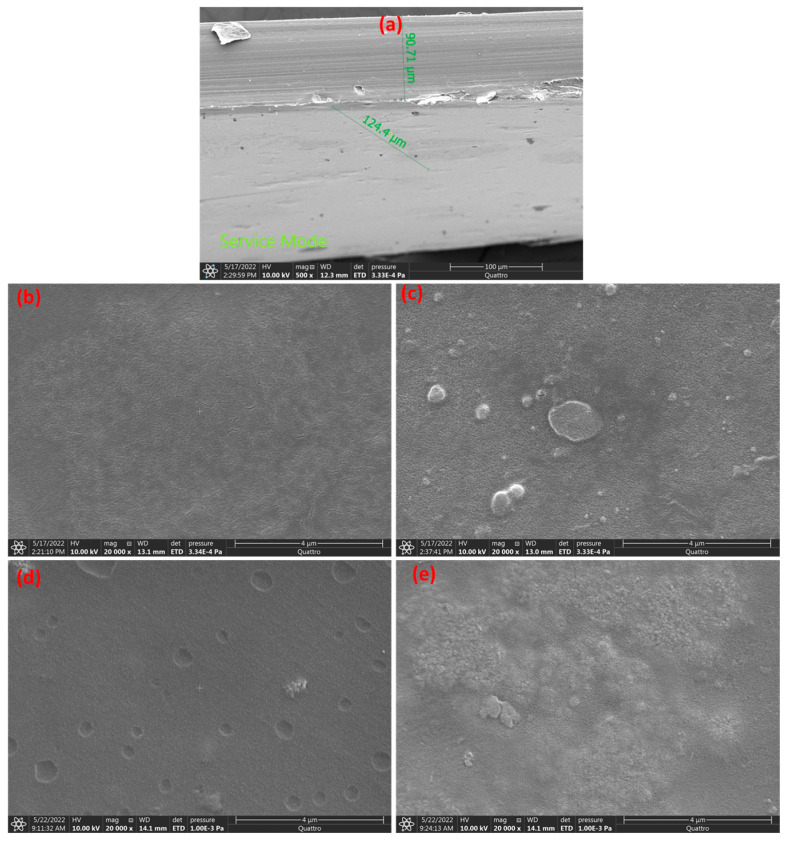
SEM of (**a**) PVA/Cs, (**b**) PVA/Cs/0.01TiO_2_, (**c**) PVA/Cs/0.025TiO_2_, (**d**) PVA/Cs/0.05TiO_2_, and (**e**) PVA/Cs/0.1TiO_2_.

**Figure 6 polymers-15-03067-f006:**
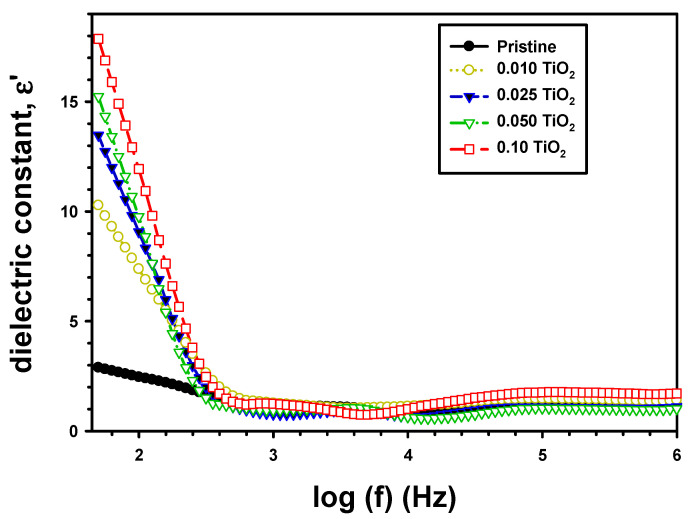
$\varepsilon^{\prime }$ with frequency for the pristine PVA/Cs and the composite PVA/Cs/TiO_2_ films.

**Figure 7 polymers-15-03067-f007:**
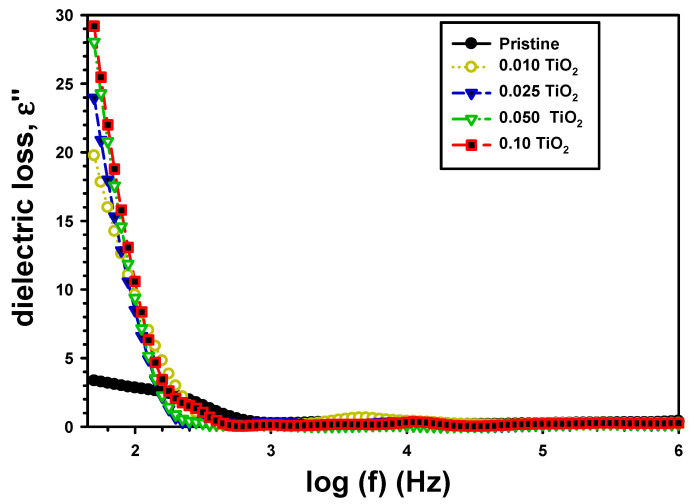
$\varepsilon^{\prime \prime }$ with the frequency of the pristine PVA/Cs and the composite PVA/Cs/TiO_2_ films.

**Figure 8 polymers-15-03067-f008:**
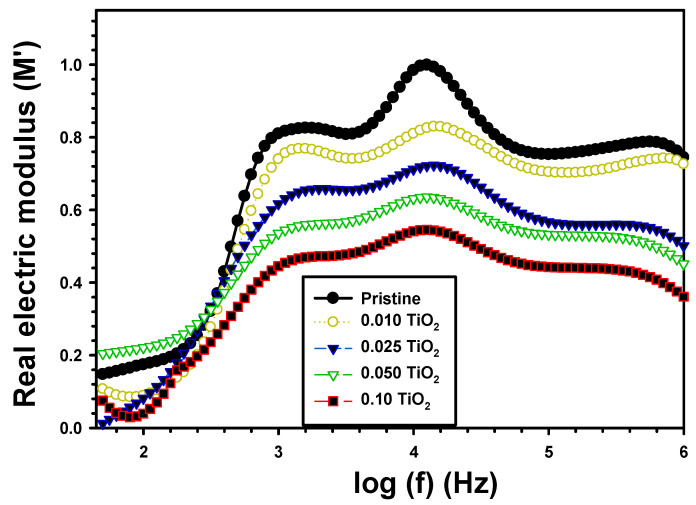
$M^{\prime }$ with frequency for the pristine PVA/Cs and the composite PVA/Cs/TiO_2_ films.

**Figure 9 polymers-15-03067-f009:**
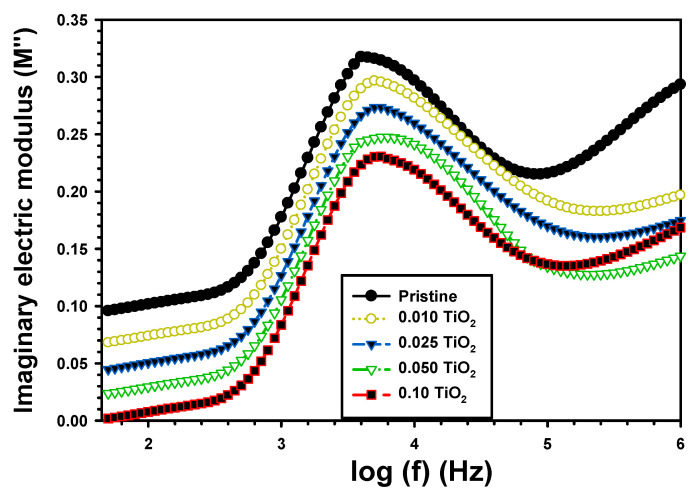
$M^{\prime \prime }$ vs. frequency of the pristine PVA/Cs and the composite PVA/Cs/TiO_2_.

**Figure 10 polymers-15-03067-f010:**
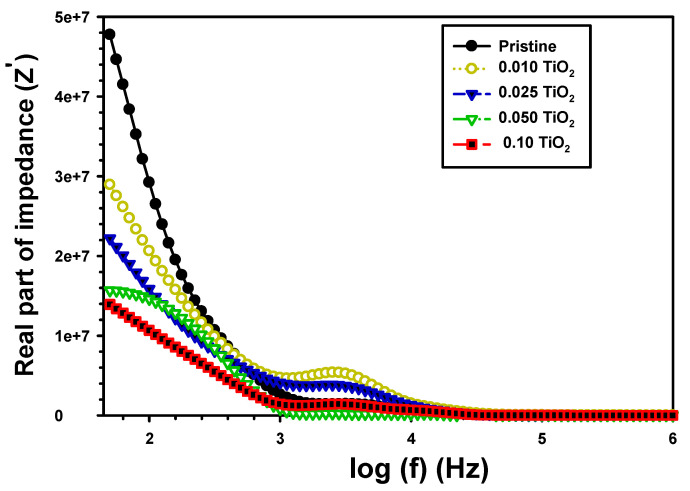
The $Z^{\prime }$ vs. frequency of the pristine PVA/Cs and the composite PVA/Cs/TiO_2_.

**Figure 11 polymers-15-03067-f011:**
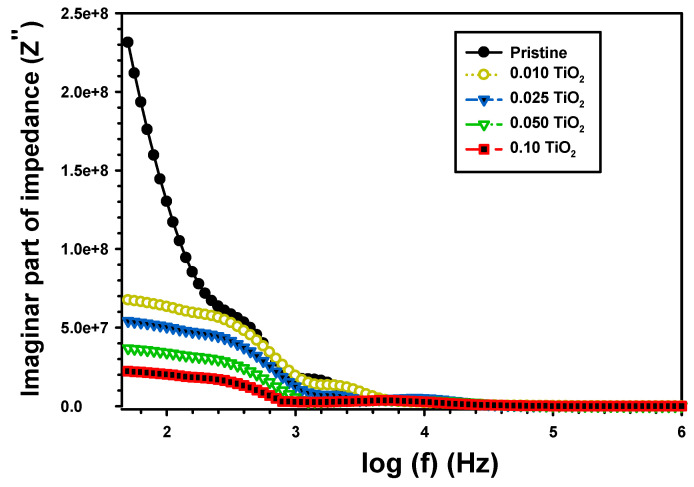
The $Z^{\prime \prime }$ vs. frequency of the pristine PVA/Cs and the composite PVA/Cs/TiO_2_.

**Figure 12 polymers-15-03067-f012:**
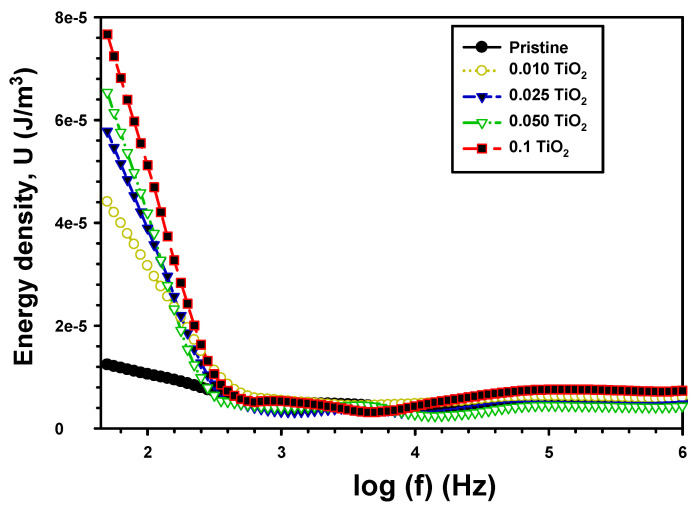
The U with frequency for the pristine PVA/Cs and the composite PVA/Cs/TiO_2_.

**Figure 13 polymers-15-03067-f013:**
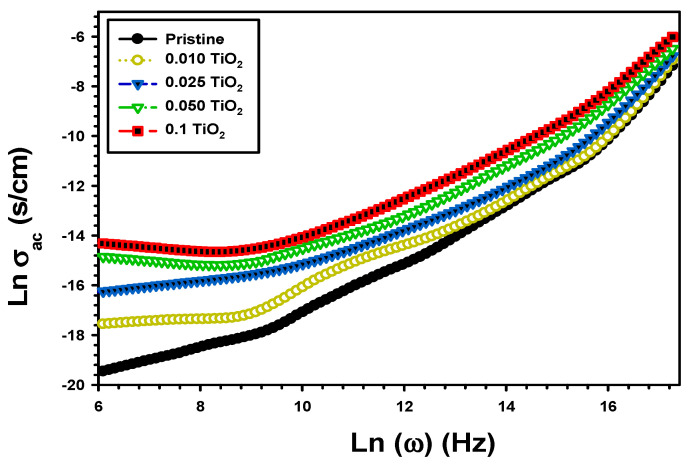
The **σ_ac_** vs. frequency of the pristine PVA/Cs and the composite PVA/Cs/TiO_2_.

**Figure 14 polymers-15-03067-f014:**
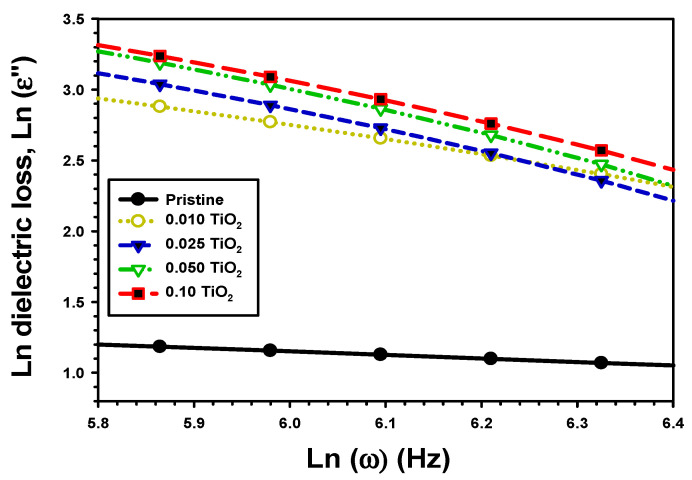
Ln $\varepsilon^{\prime \prime }$ vs. Ln (ω) for the pristine PVA/Cs and the composite PVA/Cs/TiO_2_.

**Table 1 polymers-15-03067-t001:** The $\varepsilon^{\prime }$, ε″, ***M′***, ***M″***, Z′, **σ_ac_,** and U of PVA/Cs and PVA/Cs/TiO_2_ films.

	$\varepsilon^{\prime }$	ε″	Z’	Z”	*M* *′*	*M* *″*	σ_ac_ (S/cm)	U (J/m^3^)
PVA/Cs	2.46	2.83	2.65 × 10^7^	13 × 10^7^	0.174	0.101	0.04 × 10^−7^	1.05 × 10^−5^
0.01%TiO_2_	7.38	9.61	2.07 × 10^7^	6.5 × 10^7^	0.089	0.073	0.25 × 10^−7^	3.16 × 10^−5^
0.025%TiO_2_	9.07	8.48	1.59 × 10^7^	5.2 × 10^7^	0.081	0.050	0.96 × 10^−7^	3.89 × 10^−5^
0.050%TiO_2_	9.76	9.37	1.45 × 10^7^	3.6 × 10^7^	0.221	0.029	3.38 × 10^−7^	4.18 × 10^−5^
0.1%TiO_2_	11.93	10.58	1.06 × 10^7^	2.3 × 10^7^	0.039	0.007	5.75 × 10^−7^	5.12 × 10^-^

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
