# Peer review of "Fabrication, Structural Properties, and Electrical Characterization of Polymer Nanocomposite Materials for Dielectric Applications"

_polymers, 2023, doi:10.3390/polym15143067_

Round 1

Reviewer 1 Report

1.    The current version of manuscript is like a draft instead of a final version. There are too many typos, the author really needs to improve the writing.

2.    The author should improve the logic and writing of introduction, there is no connection between each paragraph, the author just gives a brief introduction of PVA, chitosan and TiO2. More literature for PVA/TiO2 composites or polymer TiO2 nanocomposites should be reviewed.

3.    The author should point out why use PVA/chitosan as matrix? What is the role of chitosan in composites? And how does the chitosan influence the properties of PVA/TiO2 composites?

4.    There is no mechanical testing, how could the author define the film as flexible in their conclusion?

5.    The authors should point out the impact of their studies against other known arts in the literature.

The current version of manuscript is like a draft instead of a final version. There are too many typos, the author really needs to improve the writing.

Author Response

Manuscript Title: Fabrication, structural properties and electrical characterization of polymer nanocomposite materials for dielectric applications

‏‏

The author would like to thanks the editor and the reviewer for careful and thorough reading of this manuscript and for the thoughtful comments and constructive suggestions, which help to improve the quality of this manuscript. Our response follows (the reviewers comments are in black, while the author’s response is in highlight color).

Author's Reply to the Review Report (Reviewer 1)

  1. The current version of manuscript is like a draft instead of a final version. There are too many typos; the author really needs to improve the writing.

Thank you for your observation, now the current version of manuscript is improved and the language and especially grammar is completely edited by the English language editor to be more clearly (I attached the Certificate of Editing with the revision- and also attached the Track Changes on the supplementary file)

  1. The author should improve the logic and writing of introduction, there is no connection between each paragraph, the author just gives a brief introduction of PVA, chitosan and TiO2. More literature for PVA/TiO2 composites or polymer TiO2 nanocomposites should be reviewed.

Done, now the introduction is improved and is being more connection between the paragraphs.  Beside brief introduction of using PVA, chitosan and TiO2 are inserted in the text. Moreover, more literature for PVA/TiO2 composites is reviewed (Introduction section, Page 2/1st Paragraph/ line 46, Page 2/2nd Paragraph/ line 57, Page 2/3rd Paragraph/ line 73, Page 2/4th Paragraph/ line 97 )

  1. The author should point out why use PVA/chitosan as matrix? What is the role of chitosan in composites? And how does the chitosan influence the properties of PVA/TiO2 composites?

Done, now the reason for using the PVA/chitosan as matrix, the role of chitosan in composites and how does the chitosan influence the properties of PVA/TiO2 composites are mentioned in the text (Introduction section, Page 2, 2nd Paragraph, line 73)

  1. There is no mechanical testing, how could the author define the film as flexible in their conclusion?

Thank you for your observation, now removed flexible from the text. Moreover, from several studies highlight to prepare flexible composites by incorporations TiO2 on the PVA/Cs using solution-casting technique. The Incorporating TiO2 nanotubes, during the fabrication may provide a better surface area, total pore volume, and mechanical properties.

  1. The authors should point out the impact of their studies against other known arts in the literature.

Done, now the impact of their studies against other known is compare with previous studies (Introduction section, Page 2, 1st Paragraph line 46 and cited with Refs No 8.9.10.11)

Comments on the Quality of English Language

The current version of manuscript is like a draft instead of a final version. There are too many typos, the author really needs to improve the writing.

Thank you for your observation, now the current version of manuscript is improved and the language and especially grammar is completely edited by the English language editor to be more clearly (I attached the Certificate of Editing with the revision- and also attached the Track Changes on the supplementary file)

Reviewer 2 Report

The authors prepared PVA/Chitosan films with different amounts of TiO2 additive, and the films were analyzed using XRD, FTIR, EDX, SEM, and AFM. In addition, the electrical characteristics of films were comprehensively investigated. However, there are some additional issues that need to be addressed before publication.

1.    
The full name should provide when it was mentioned at the first time. For example, XRD, FTIR, SEM, AFM, and EDX in the abstract.

2.     The authors should carefully check the whole manuscript, there are many mistakes. For example, 0.0.25×10-7 s cm-1 in the abstract.

3.     Please elaborate on the statement that "the physical and chemical characteristics of PVA are limited" by providing a detailed explanation of the specific limitations.

4.     While the authors have emphasized the advantages of Chitosan in life science, it is crucial to discuss its advantages in the electrochemical field as well.

5.     Please focus on highlighting the merits of TiO2 specifically in the electrochemical field, elucidating its beneficial properties and applications.

6.     Please provide comprehensive characterization of TiO2, including techniques such as SEM and XRD analysis, to offer a detailed understanding of its structure and morphology.

7.     Please rectify the incorrect labels in Fig. 4a and Fig. 5d to accurately represent the experimental results.

8.     Please discuss the advantages resulting from increased roughness and hydrophilicity in the PVA/Chitosan blend covered by TiO2, addressing their significance in the context of the study's objectives.

9.     What’s the reason for the smoother surface of PVA/Cs/0.05TiO2 as compared to PVA/Cs/0.01TiO2 ?

10.  Please determine the optimal sample based on the defined criteria and evaluation parameters, considering the desired properties and performance requirements.

11.  What’s the impact of different ratios of PVA and Chitosan on the electrical characteristics of the sample?

The format problems and English Language in this paper need to pay more attention. The font size of in Figure 1, for instance, is inconsistent.

Author Response

Manuscript Title: Fabrication, structural properties and electrical characterization of polymer nanocomposite materials for dielectric applications

Author's Reply to the Review Report (Reviewer 2)

Comments and Suggestions for Authors

The authors prepared PVA/Chitosan films with different amounts of TiO2 additive, and the films were analyzed using XRD, FTIR, EDX, SEM, and AFM. In addition, the electrical characteristics of films were comprehensively investigated. However, there are some additional issues that need to be addressed before publication.

  1. The full name should provide when it was mentioned at the first time. For example, XRD, FTIR, SEM, AFM, and EDX in the abstract.

Thank you for your observation, now the full name of  XRD, FTIR, SEM, AFM, AFM and EDX is provided in the abstract ((Abstract section, Page1/ line 20 )

  1. The authors should carefully check the whole manuscript, there are many mistakes. For example, 0.0.25×10-7 s cm-1 in the abstract.

Thank you for your observation, now the current version of manuscript is revised and rechecked. Moreover the language and especially grammar is completely edited by the English language editor.

  1. Please elaborate on the statement "the physical and chemical characteristics of PVA are limited" by providing a detailed explanation of the specific limitations.

Thank you for your comment, now more explanation is provided in the manuscript to show the addition of Cs and TiO2 nanoparticles is improved the physical/chemical properties of the composite (Introduction section, Page 2/3rd Paragraph/ line 73, Page 2/4th Paragraph/ line 97 )

  1. While the authors have emphasized the advantages of Chitosan in life science, it is crucial to discuss its advantages in the electrochemical field as well.

Thank you for your observation, now the advantages of chitosan in the electrochemical version of manuscript is emphasized in the manuscript. (Introduction section, Page 2/2nd Paragraph/ line 57)

  1. Please focus on highlighting the merits of TiO2specifically in the electrochemical field, elucidating its beneficial properties and applications.

Thank you for your observation, now the advantages of TiO2 in the electrochemical version of manuscript is emphasize in the manuscript. (Introduction section, Page 2/4th Paragraph/ line 97 )

  1. Please rectify the incorrect labels in Fig. 4a and Fig. 5d to accurately represent the experimental results.

Thank you for your observation, now the labels in Fig. 4a and Fig. 5d is corrected

  1. Please discuss the advantages resulting from increased roughness and hydrophilicity in the PVA/Chitosan blend covered by TiO2, addressing their significance in the context of the study's objectives.

Thank you for your observation, now the advantages resulting from increased roughness and hydrophilicity in the PVA/Chitosan blend covered by TiO2 is mentioned in the objective work. . (Introduction section, Page 3/2nd  Paragraph/ line 102 )

  1. What’s the reason for the smoother surface of PVA/Cs/0.05TiO2as compared to PVA/Cs/0.01TiO2 ?

Thank you, now the reason for the smoother surface of PVA/Cs/0.05TiO2 as compared to PVA/Cs/0.01TiO2 is discussed in the text (Result section, Page 6, 1st  Paragraph line 178).

  1. Please determine the optimal sample based on the defined criteria and evaluation parameters, considering the desired properties and performance requirements.

Thank you for your observation, now optimal sample, evaluation parameters, and performance requirements is discussed in the manuscript. . (Experimental section, Page 3/2nd Paragraph/ line 112, Line 126 )

  1. What is the impact of different ratios of PVA and Chitosan on the electrical characteristics of the sample?

Thank you for your observation, now the impact of different ratios of PVA and Chitosan on the electrical characteristics is discussed in the manuscript. . (Result section, Page 8/ 2nd Paragraph/ line 219, Result section, Page 8/ 3rd  Paragraph/ line 240, Result section, Page 14/ 2nd Paragraph/ line 346 )

Reviewer 3 Report

The manuscript entitled “Fabrication, structural properties and electrical characterization of polymer nanocomposite materials for dielectric applications” was submitted by authors. The article is nicely written and presented since the study's topic fits the journal's scope. I would recommend a thorough revision before it can be considered for publication. Please, carefully read my considerations in your conclusions.

1.      Language editing is necessary.

2.      Methodology part should be improved.

3.      Rewrite the line 91-97.

4.      There are two 5 d pics. Provide 5e pic.

5.      Improve the FT-IR results and discussion.

6.      Typological errors –rectify.

7.      Revise the keywords.

8.      There are two 4 b pics. Provide 4a pic.

9.      AFM for e) PVA/chitosan/0.1TiO2 films not provided.

10.  Fig.4: AFM of (a) pristine PVA/chitosan and the composite (b) PVA/chi- 146 tosan/0.025TiO2,(d) PVA/chitosan/0.05TiO2 and (e) PVA/chitosan/0.1TiO2 films. I hope ‘c’ is missing.

The manuscript entitled “Fabrication, structural properties and electrical characterization of polymer nanocomposite materials for dielectric applications” was submitted by authors. The article is nicely written and presented since the study's topic fits the journal's scope. I would recommend a thorough revision before it can be considered for publication. Please, carefully read my considerations in your conclusions.

1.      Language editing is necessary.

2.      Methodology part should be improved.

3.      Rewrite the line 91-97.

4.      There are two 5 d pics. Provide 5e pic.

5.      Improve the FT-IR results and discussion.

6.      Typological errors –rectify.

7.      Revise the keywords.

8.      There are two 4 b pics. Provide 4a pic.

9.      AFM for e) PVA/chitosan/0.1TiO2 films not provided.

10.  Fig.4: AFM of (a) pristine PVA/chitosan and the composite (b) PVA/chi- 146 tosan/0.025TiO2,(d) PVA/chitosan/0.05TiO2 and (e) PVA/chitosan/0.1TiO2 films. I hope ‘c’ is missing.

Author Response

Manuscript Title: Fabrication, structural properties and electrical characterization of polymer nanocomposite materials for dielectric applications

Author's Reply to the Review Report (Reviewer 3)

Comments and Suggestions for Authors

The manuscript entitled “Fabrication, structural properties and electrical characterization of polymer nanocomposite materials for dielectric applications” was submitted by authors. The article is nicely written and presented since the study's topic fits the journal's scope. I would recommend a thorough revision before it can be considered for publication. Please, carefully read my considerations in your conclusions.

  1. Language editing is necessary.

Done, now the language and especially grammar is completely edited by the English language editor to be more clearly (I attached the Certificate of Editing with the revision- and also attached the Track Changes on the supplementary file)

  1. Methodology part should be improved.

Thank you for your observation, now the Methodology part is modified (Experimental section, Page 3/2nd Paragraph/ line 112, Page 3/3rd Paragraph/ line 126)

  1. Rewrite the line 91-97.

Thank you for your observation, now line 91-97 were rewritten (Page 3/3rd Paragraph/ line 126-131)

  1. There are two 5 d pics. Provide 5e pic.

Thank you for your observation, now replace 5d with 5e in Figure 5 ( Page 7 /line 199)

  1. Improve the FT-IR results and discussion.

Done, now the discussion of FT-IR results were improved and mentioned in the text (Page 4 /line 157)

  1. Typological errors –rectify.

Thank you for your observation, now the Typological errors  were revised

  1. Revise the keywords.

Done, now the keywords were revised ( Page 1 /line 33)

  1. There are two 4 b pics. Provide 4a pic.

Thank you for your observation, now is replaced 4b with 4a in Figure 4 (Page 6 /line 185 )

  1. AFM for e) PVA/chitosan/0.1TiO2 films not provided.

Thank you for your observation, now PVA/chitosan/0.1TiO2 films is provided (Page 6 /line 185 )

  1. Fig.4: AFM of (a) pristine PVA/chitosan and the composite (b) PVA/chitosan/0.025TiO2,(d) PVA/chitosan/0.05TiO2 and (e) PVA/chitosan/0.1TiO2 films. I hope ‘c’ is missing.

Thank you for your observation, now is Fig.4: AFM revised and modified (Page 6 /line 185)

Round 2

Reviewer 1 Report

There are still many typo in the current version. For example, page1. line9. and line 14 'Egyp'. The author really revise the manuscript carefully.

'Flexible' still show in abstract.

Author Response

Reviewer comments: There are still many typo in the current version. For example, page1. line9. and line 14 'Egyp'. The author really revise the manuscript carefully. 'Flexible' still show in abstract.

Authors response: Thank you for your observation, now the typo and errors of the current version were carefully revised in the full manuscript. 

Reviewer 2 Report

No comments

Author Response

Reviewer comments: No Comments

Authors Response: Thank you